# Monitoring Chemical Accidents in Industrial Complexes Using Tower-Installed Infrared System for Remote Chemical Detection and Long-Range Video Surveillance System

**Seul Gi Lee, Eun Hee Kim and Byung Chol Ma \***

Department of Chemical Engineering, Chonnam National University, 77 Yongbong-ro, Buk-gu, Gwangju 61186, Republic of Korea
**\*** Correspondence: anjeon@jnu.ac.kr

**Abstract:** Chemical industrial complexes are extensive, complex structures with large-scale chemical facilities where large quantities of various chemical substances are handled. Detection equipment must be installed in high locations to monitor these industrial complexes and detect chemical accidents from a distance. In previous studies, individual monitoring equipment was temporarily installed on the ground, on a rooftop, or on a vehicle to detect chemical accidents from a distance. In this study, however, the industrial complex chemical accident monitoring system was developed by combining different technologies and was installed on a tower. For the Yeosu National Industrial Complex (which functioned as a test bed), 70m-high steel towers were built. Additionally, an infrared system for remote chemical detection (SIGIS-2, Bruker) and a long-range video surveillance system (TORUSS-LR2000, Globalsystems) were installed at the top of steel towers to monitor the entire industrial complex. The target substances to be monitored in real time by the infrared system for remote chemical detection were selected, and the monitoring sections were classified to enable each piece of equipment to distinguish the scanned areas. To improve the accuracy of the detection results, the information about the actual handled substances and respective facilities of the sites in the industrial complex was inserted into the database of the system and then connected to the sections. During the three-month test operation, various chemical substances (including 1,3-butadiene, methanol, methylamine, ethyl acetate, ammonia, and vinyl chloride) were detected at each section in 20,034 cases, and the detection results were consistent with the inserted actual information. The accumulated detection data shows that the detection frequency of a specific chemical substance was high in each section. This can be used as a basis for modifying the threshold of the anomaly detection model, thereby improving the accuracy of the system. Therefore, this system can detect and evaluate the leakage of chemical substances and the occurrence of fires or smoke through large-scale scans 24 hours per day. Furthermore, it can be used for the early detection of and effective responses to chemical accidents in industrial complexes.

**Keywords:** remote monitoring; chemical accident; infrared system; remote chemical detection; video surveillance system

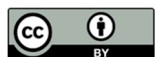

## 1. Introduction

Chemical accidents (including chemical leakages, fires, and explosions) are prone to occur in industrial complexes where chemical substances are being handled. In South Korea, 711 chemical accidents occurred from January 2014 to September 2022, including 567 chemical leakages (approximately 80% of the total number of accidents) and 103 explosions (approximately 14%) [1]. For chemical accidents, risk reduction, and effective response to accidents, a monitoring system should be installed for each industrial complex to enable early detection and identification of an accident. However, industrial complexes

are extensive and complex structures with large-scale chemical facilities where various chemical substances are handled. Thus, a monitoring system must comprise detection equipment in high locations to detect chemical accidents from a distance.

In the event of chemical leakages, it is necessary to obtain real-time information on the identified toxic compounds and their concentrations and to identify the gas source and affected areas through images showing the location and size of the toxic gas cloud [2]. Among the different techniques, passive remote sensing technology with Fourier transform infrared (FTIR) spectroscopy is suitable for large-scale monitoring to detect and identify the gas clouds in the atmosphere [3–5]. These techniques are applied in various fields and studies where the composition of volcanic gases are measured remotely (e.g., $SO_2$ detection at 6 km in New Zealand in 1996; $SO_2$, and $SiF_4$ detection at 17 km in Mexico in 1997 [6]), the emissions from industrial facilities are analyzed [4,7,8], and the atmospheric environment is in analyzed [5,9–11]. In the event of a fire, the areas where the chemical accident has occurred must be swiftly identified while observing the daytime or nighttime conditions of the industrial complex via real-time transmission of images. Long-range cameras and real-time video streaming with electro-optical (EO) and infrared (IR) systems enable the detection, recognition, and identification (DRI) of threats/targets from a distance; the warning signals are sufficiently fast for response actions [12]. Various researchers have used EO/IR sensors for different purposes, including the determination of the direction of a forest fire from smoke and heat signals [13], marine search and rescue [14], remote sensing of an air vehicle [15], and aerial monitoring of offshore oil spills [16]. Remote sensing change detection is a process used to identify changes by remote sensing images in various fields (including disaster assessment, urban studies, and environmental monitoring). A fuzzy topology-based majority voting model [17] and a novel higher-order clique conditional random field model [18] can be applied to unsupervised change detection. Recent studies related to environmental monitoring have used different detection technologies. The EU-SENSE is a system for real-time detection, monitoring, and analysis of chemical, biological, radiological, and nuclear threats and hazards. It was designed and developed as a heterogeneous sensor node that combines ion mobility spectrometry, flame photometry, and a metal oxide detector prototype [19]. In another case, a drone with sensors (photo ionization detectors and ion mobility spectrometry) and a sampling system is triggered by an alarm from a laser-based network when anomalies are detected. It is used for environmental monitoring and detects emission sources from industrial releases and forest fires [20].

In previous studies, individual pieces of equipment (including gas detectors and surveillance cameras) were temporarily installed on the ground, on rooftops, or on vehicles to detect chemical accidents from a distance, but one detection result alone was insufficient to determine chemical accidents, and it was difficult to identify the detection trend. Thus, this study presents a practical system for monitoring chemical accidents. Different technologies were combined (i.e., remote chemical detection, remote video surveillance, industrial complex information, and an artificial intelligence (AI) chemical accident detection model) to develop the industrial complex chemical accident monitoring system. The goal is the early detection of chemical accidents that may occur in industrial complexes through large-scale scans; the system can detect and evaluate chemical leakages and fires or smoke. The industrial complex chemical accident monitoring system was applied to the Yeosu National Industrial Complex (YNIC), located in Jeollanam-do. Since the chemical factories of the YNIC are densely packed without sufficient safe distance from other business sites, accidents, such as chemical leakages, will likely affect the neighboring companies and residents [21]. To create the chemical accident monitoring system, 70m-high steel towers were built and installed at two locations (Jeongnyang and Hwachi) within the YNIC. Moreover, an IR system for remote chemical detection (IR-RCD) and a long-range video surveillance system (LRVS) were installed at the top of each tower to enable the system to monitor the entire industrial complex. The total area of the industrial complex was divided into 50 m × 50 m cells, and coordinates and numbers were then assigned

to each cell. Subsequently, the monitoring area was divided into 12 ° sections, and the respective cell numbers were assigned to each section. Thus, the cell number is the key to linking information to a specific area in the system. To detect chemicals remotely, substances that are handled in the YNIC, as well as those that are detectable in real time by the IR-RCD, were selected as targets for monitoring. Thereafter, the latitude and longitude information of the facilities handling the target substances was identified and linked to the cell numbers. The IR-RCD and the LRVS each generate chemical accident alerts after analyzing the actual chemical handling information in the section corresponding to a detected anomaly while scanning the industrial complex. These chemical accident alerts will enable the persons in charge to respond more swiftly by identifying the type of chemical accident and its approximate location. The main contributions of this article include the following:

(1) This system can detect and evaluate the leakage of chemical substances and the occurrence of fires or smoke through large-scale scans.

(2) The system can be used for early detection of and effective responses to chemical accidents in industrial complexes.

(3) The information on chemical substances detected in each section can be used to identify routine emissions of substances from workplaces.

(4) Based on this, the government can manage and supervise workplaces in industrial complexes.

## 2. Methods

The industrial complex chemical accident monitoring system consists of chemical accident recognition equipment, facilities, industrial complex information, and software solutions. Each element is interconnected to send and receive signals for operating the system. Figure 1 shows the overall configuration of the industrial complex chemical accident monitoring system.

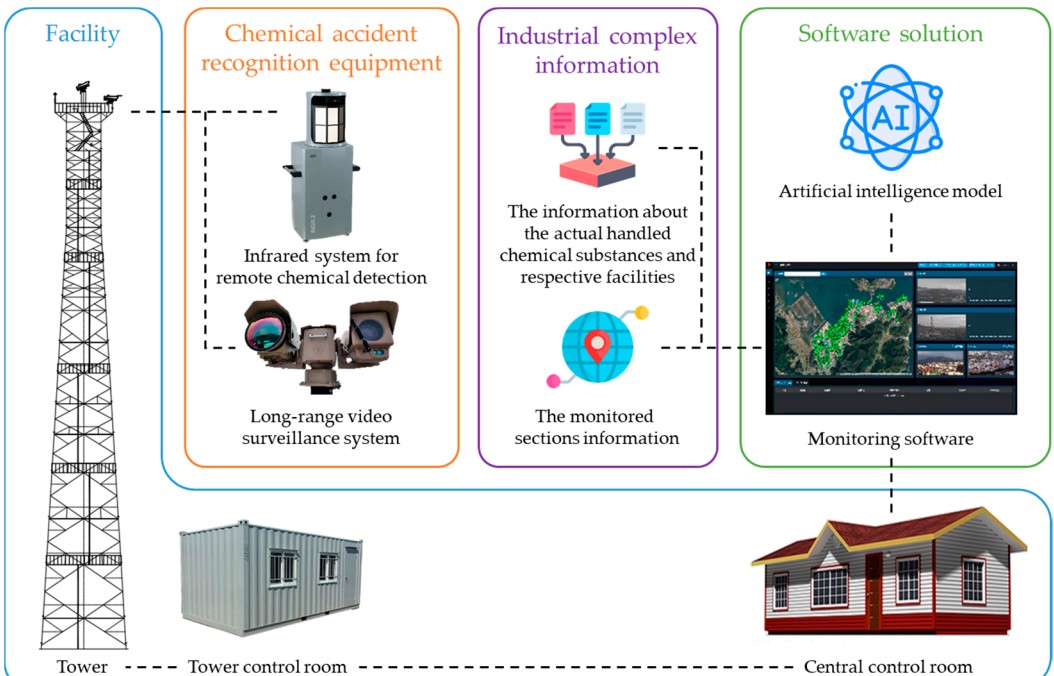

**Figure 1.** Overall configuration of the industrial complex chemical accident monitoring system.

### 2.1. The Chemical Accident Recognition Equipment

The chemical accident recognition equipment can scan the entire area of an industrial complex and determine whether chemical leakages, fires, or smoke have occurred. The

Scanning Infrared Gas Imaging System (SIGIS-2, Bruker, Germany) was used as the IR-RCD. Moreover, the Tactical Observer & Radar Unified Surveillance System (TORUSS-LR2000, Globalsystems, South Korea), which provides visible and thermal images of industrial complexes and measures the distance between points, was used as the LRVS. Each type of chemical accident recognition equipment was operated using dedicated software, and the detected or observed data were transmitted to the main server.

2.1.1. Infrared System for Remote Chemical Detection (SIGIS-2)

The SIGIS-2 was selected due to the type of target substances, detectable distance, and sensitivity of the IR-RCD. The SIGIS-2 is based on IR spectroscopy and comprises a single-element detector and a scanning mirror. The interferometer is a Michelson, which has a spectral resolution of approximately 0.5 cm$^{-1}$ and provides the capability to distinguish chemical compounds with similar spectra; the spectral range of the SIGIS-2 is 680 cm$^{-1}$ to 1500 cm$^{-1}$, and it includes a spectral library of 450 or more industrial chemical compounds [2,22]. Thus, this system, which can track particle phases and detect low concentrations, can assist in the prompt monitoring of chemical leakages. This system sequentially measures the selected area using a scanning mirror and automatically analyzes the IR spectrum (in real time), which is then visualized by overlaying a chemical image onto a video image. The analysis algorithm of the SIGIS-2 calculates the coefficient of correlation between the measured spectrum and a reference spectrum. A chemical is identified as such if the coefficient of correlation and the measured signal intensity are above predefined thresholds. This process makes it possible to verify the identified chemical, as well as the position and size of a gas cloud [2,22].

Figure 2 shows the identification image and the coefficient of correlation image for an instance when ammonia was detected by the SIGIS-2 installed at the Jeongnyang site. For the region where ammonia has been identified, the coefficient of correlation is nearly white, i.e., it was close to 1.

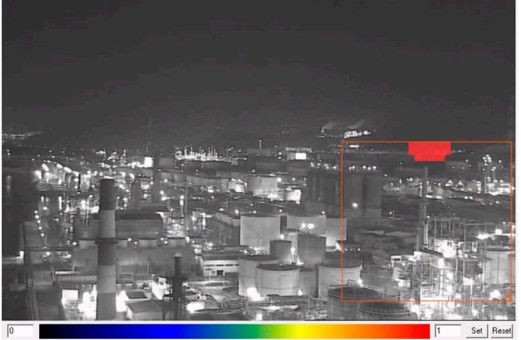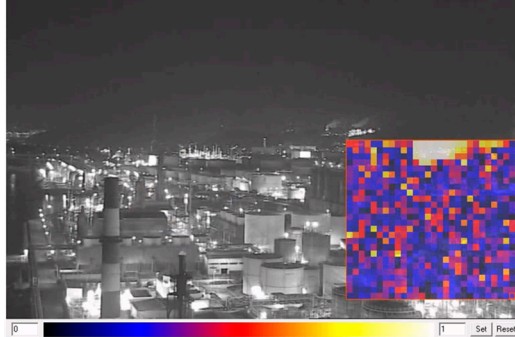

**Figure 2.** The identification image (left) and the coefficient of correlation image (right) for the instance when ammonia was detected by the SIGIS-2 installed at the Jeongnyang site (Date: 21 Sep 2022 at 22:57 GMT+9).

In the industrial complex chemical accident monitoring system, the SIGIS-2 program mode was set to "automatic and repeated scanning" for the entire industrial complex area. The head angle and array size were set mainly based on the location of the chemical facility in each section, which was divided by approximately 12°, and camera rotation was minimized for rapid scanning. The scan time per unit section, set to 12° for pan and 30° for tilt, is about 1 minute. Figure 3 illustrates the program mode of the SIGIS-2 installed at the Jeongnyang site; a panorama was constructed by connecting the recorded sections (marked with orange squares in Figure 3).

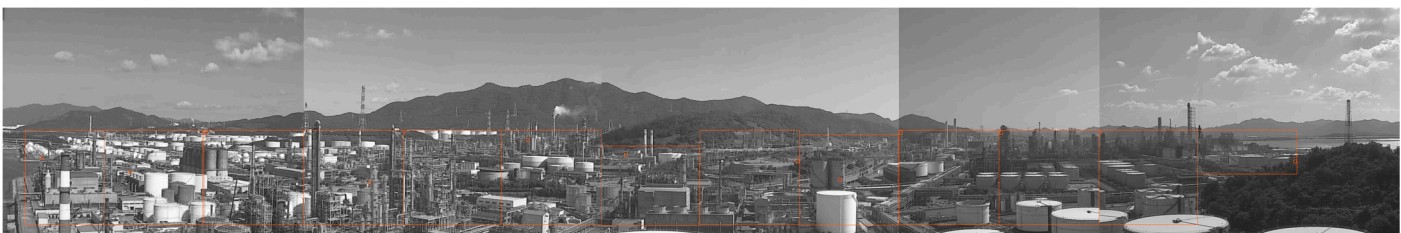

**Figure 3.** Program mode of the SIGIS-2 installed at the Jeongnyang site.

The SIGIS-2 scans the industrial complex in the order specified in the program mode; it simultaneously stores the optical camera images, chemical detection images, and original data by section in the network-attached storage (NAS). It shares the analysis of the results of the identified chemicals with the control software. Moreover, a function was developed to detect chemicals by rotating the SIGIS-2 to the corresponding section and scanning it when the LRVS or the AI model detects a chemical accident.

Thus, the IR-RCD (SIGIS-2), can detect and identify chemical leakages in real time. The shape of a gas cloud and the position of the accident are verified using meteorological data (temperature and wind direction). The meteorological data is measured every minute using a separately installed meteorological instrument and transmitted to the main server. Furthermore, the equipment can recognize chemical substances that are routinely generated in each section within the industrial complex through the accumulated observation data; it uses this information to improve the detection accuracy for chemical accidents.

### 2.1.2. Long-Range Video Surveillance System (TORUSS-LR2000)

The LRVS is composed of EO/IR multi-sensors, which can provide accurate information during both daytime and nighttime because the EO sensor converts light into an electrical signal, and the IR sensor detects surrounding structures by emitting or sensing IR radiation. Thus, the LRVS is mainly used to improve target identification, evaluate threats at a certain distance, and monitor targets. Owing to its excellent image stabilization and long-range imaging ability, this system is used for national defense and commercial purposes. It can be applied in different sectors, such as situational awareness evaluation, fire control systems, aviation homeland security, search and rescue, and intelligence, surveillance, and reconnaissance [23].

The industrial complex chemical accident monitoring system was developed to monitor chemical accidents from a distance; thus, it must be able to accurately identify distant targets. The TORUSS-LR2000 was applied because of its suitable DRI characteristics, which indicate the degree to which a camera can generate an image of a specific target. Table 1 shows the DRI characteristics of the TORUSS-LR2000.

**Table 1.** DRI characteristics of the TORUSS-LR2000.

| Type | Detection | Recognition | Identification |
|------|-----------|-------------|----------------|
| EO camera | 19.8 km | 9.3 km | 6.0 km |
| IR camera | 18.8 km | 10.0 km | 5.5 km |

The TORUSS-LR2000 is an intelligent LRVS that uses long-distance multi-sensors (EO/IR) and a high-precision driving device [24]. The FLIR Ranger HDC-1200 (United States) was used as the IR camera. The Ranger HDC-1200, together with the FLIR image processing engine, provide long-range thermal images and a 22x continuous optical zoom for constant situational awareness and target evaluation [25].

The TORUSS-LR2000 outputs real or thermal images in black and white or rainbow colors. The thermal image can show gas or water vapor emitted from a stack, which is nearly invisible in a real image and to the naked eye. Figure 4 shows an image of the

TORUSS-LR2000 installed at the Hwachi site; the thermal image underneath shows the heat generated from the stack, which is invisible in the real image.

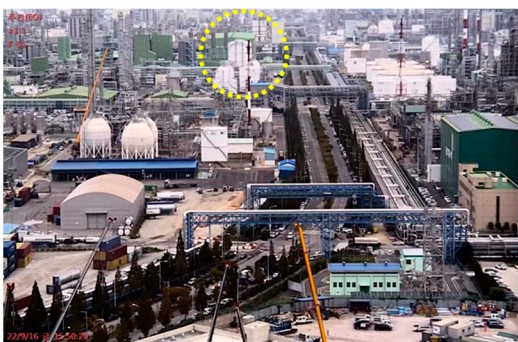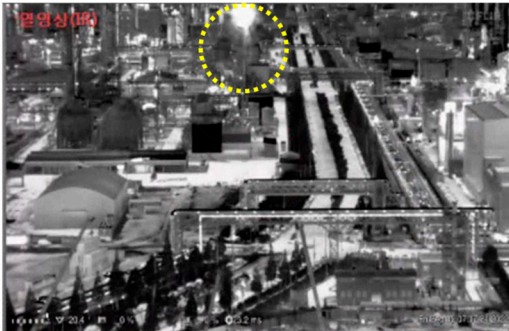

**Figure 4.** EO image (**left**) and IR image (**right**) of the TORUSS-LR2000 installed at the Hwachi site (Date: 16 Sep 2022 at 15:50 GMT+9).

The TORUSS-LR2000 is normally operated in an auto-scan mode that automatically passes through predesignated points, storing still images from each section into the NAS and sharing them with the AI chemical accident detection model. Moreover, a function that automatically directs this system to the corresponding section was developed in the event that the SIGIS-2 detects chemical substances.

Hence, the LRVS (TORUSS-LR2000) can identify targets (such as chemical facilities), recognize the shape of gas or water vapor through thermal imaging, enlarge the accident site, and measure the distance to allow location estimation.

## 2.2. The Facilities of the Industrial Complex Chemical Accident Monitoring System

The facilities for operating the industrial complex chemical accident monitoring system include a steel tower, a steel tower control room, and a central control room.

To monitor an industrial complex, the equipment should be installed at a high location to secure wide visibility. In this study, a 70m-high steel tower was installed at the location indicated in Figure 5; the chemical accident recognition equipment was installed at the top of the steel tower to monitor the entire industrial complex. The steel tower is a steel truss structure with square cross sections designed according to both international and South Korean standards [26], as shown in Table 2.

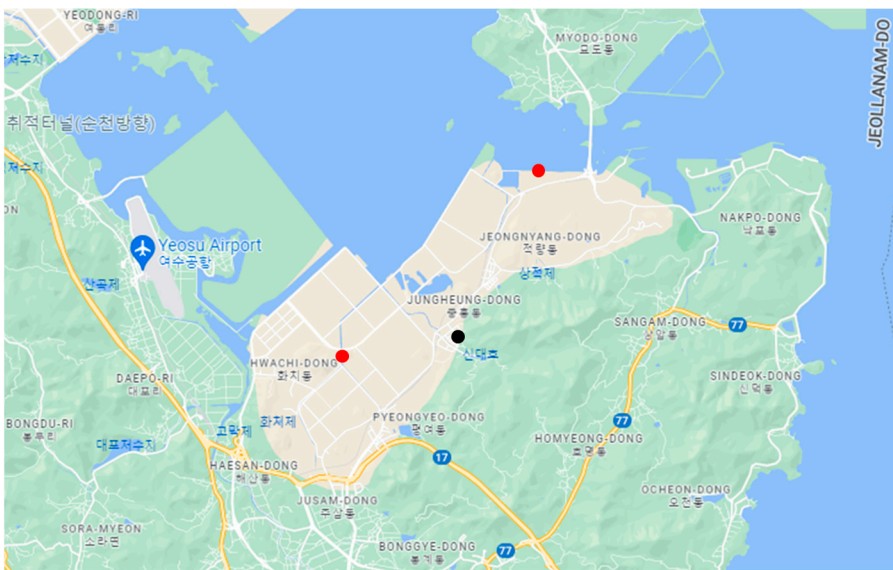

**Figure 5.** Location of the tower (red dot) and central control room (black dot). (Source: Google maps).

**Table 2.** Design characteristics of the steel tower.

| Category | Design | |
|:---:|:---:|:---:|
| Safety factor, FS | 2.0 | |
| Wind load, $W_D$ [1] | Max. 60 m/s | |
| Seismic conditions, Z [2] | Zone I | 0.22g |
| Displacement [3] | Vertical 1/1200 | Horizontal 1/800 |

[1] KBC 2016, [2] KDS 41 17 00, [3] JEC-127-1979.

Figures 6 and 7 present the chemical accident recognition equipment at the top of the steel tower with a load capacity of 500 kg (to hold the equipment and operators). To ensure a sufficient field of view of each piece of equipment, the equipment was arranged in the forward direction, and the height of the front-part handrail was lowered to 500 m. An industrial lift was used to go up and down the steel tower.

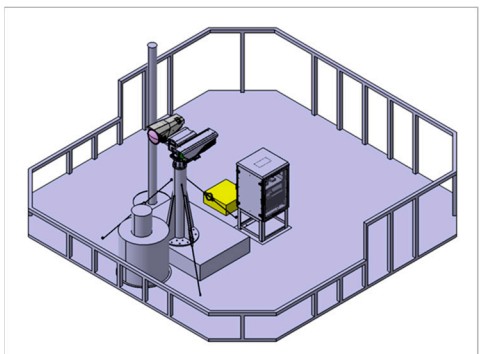 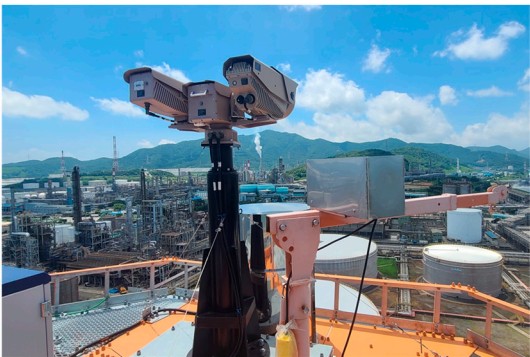

**Figure 6.** The three-dimensional (3D) drawing (**left**) and photograph (**right**) of the equipment installed on the steel tower top at the Jeongnyang site.

The steel tower control room is located right below the steel tower; it supplies the required power and telecommunication with the equipment. Since moisture, which is a strong IR absorber, can degrade the equipment's performance, a compressor was placed in the tower control room and connected to the window in front of the SIGIS-2. Moreover, a nitrogen atmosphere was created around the SIGIS-2 to prevent moisture or contaminants from entering the equipment.

The central control room is located approximately 5–8 km away from the steel tower; it is used to store and display information that has been observed or photographed by the chemical accident recognition equipment; thus, it serves as an integrated monitoring and coordination facility. The operator performs remote monitoring in real time in the central control room, i.e., he/she can check the program mode and auto-scan screens set by default for each software through a display or manually control each piece of software on a desktop PC.

### 2.3. The Industrial Complex Information

The information about chemical substances that have only been identified by the chemical accident recognition equipment (or with images) is insufficient for responding to chemical accidents. Thus, the actual chemical substances, handling facilities, and locations of the workplaces in the industrial complex are also considered in the system; the actual chemical handling information is connected to the respective section where the equipment scans in real time to increase the reliability of the detection results.

For the chemical substances handling information, the chemical substances that are handled at the site and can be detected in real-time by the IR system (SIGIS-2) were chosen as the target substances. In this study, the sites (factories) and handling facilities that manufacture or use the target substances within the industrial complex were investigated. The

handling substances, capacity, height, latitude, and longitude of each facility were determined based on the data submitted by the relevant sites (factories) to the Korean Ministry of Environment. The surveyed data were coded (F: Factory, M: Material) and inserted into the database of the chemical accident monitoring system.

Regarding the information about the monitored sections, the total area of the industrial complex was divided into 50m × 50m grid cells with respective coordinates and numbers. Next, the monitored sections were divided by 12°, and the pan and tilt of each section were adjusted to the program mode of the SIGIS-2 and the auto-scan mode of the TORUSS-LR2000. This manipulation enables each piece of equipment to distinguish the scanned areas. Thus, the ranges observed are 156° (= 13 sections × 12°) and 240° (= 20 sections × 12°) areas concerning the steel towers at the Jeongnyang and Hwachi sites, respectively. Figure 7 presents the cell number corresponding to each section. The cell numbers are the keys that connect the chemical substance handling information in the system with the information about the monitored sections.

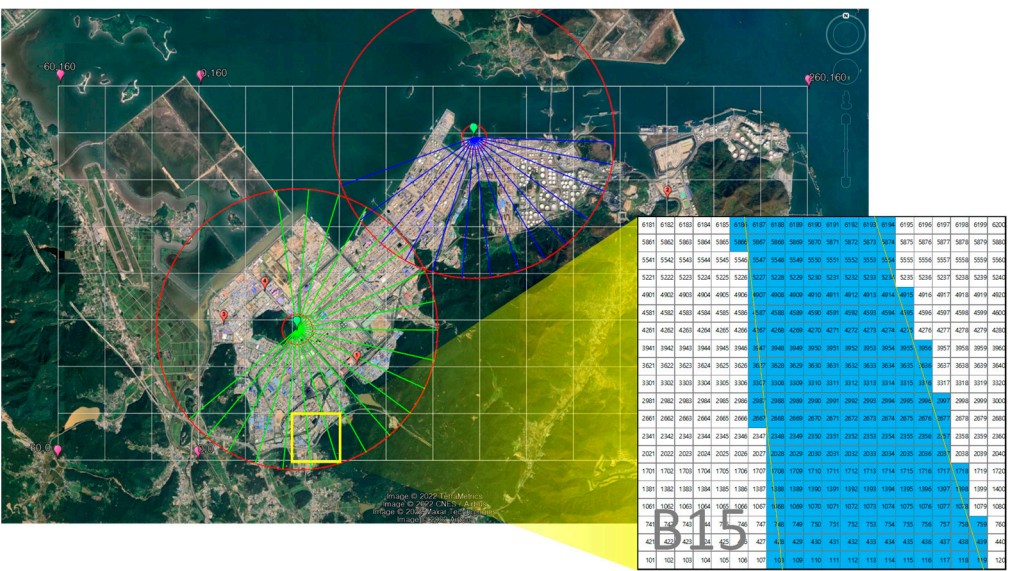

**Figure 7.** The monitored sections and sample of cell numbers corresponding to the section (B15).

The industrial complex chemical accident monitoring system generates a chemical accident alert after analyzing the observed data and actual chemical handling information. The following is an illustration of how the chemical handling information and information about the monitoring section can be connected and utilized:

- The SIGIS-2 program mode scan and chemical substance detection;
- Transmission of the information about the corresponding section and substance;
- The TORUSS-LR2000 automatic rotation to detected section;
- Verification of monitoring software: cell number of the handling facility corresponds to the section;
- Verification of workplace and substance codes of the corresponding handling facility;
- Creation of a list of workplaces in which the chemical accident may have occurred;
- Chemical accident alert.

### 2.4. The Software Solutions

The software solutions comprised the monitoring software and the AI chemical accident detection model. The monitoring software used Java and JavaScript, in HTML5 web format, and was developed based on the SPRING 4 framework. The AI monitoring server application used a flask-based Python script.

The monitoring software manages all the equipment components, observation data, errors, and chemical accident histories in a centralized manner and uses the stored industrial complex information. Figure 8(a) displays the location (marker) of the chemical handling facilities (created in advance based on Google Maps) and the section that was scanned by the SIGIS-2 in real time. Figure 8(b) displays the zones that every SIGIS-2 scans in real time; it shows the column density (CL) of the chemical analyzed by the SIGIS-2 upon detection. Figure 8(c) displays the selected EO or IR images that the TORUSS-LR2000 scans in real time. In this case, an extended module appears when the operator clicks on an image; it allows simple manipulations, such as adjustment of the camera direction, zoom, and focus. Figure 8(d) presents a list of possible chemical accident sites based on the chemical handling information. In this case, the location point (i.e., the marker) of the chemical substance handling facility at a suspected accident site changes from green to red.

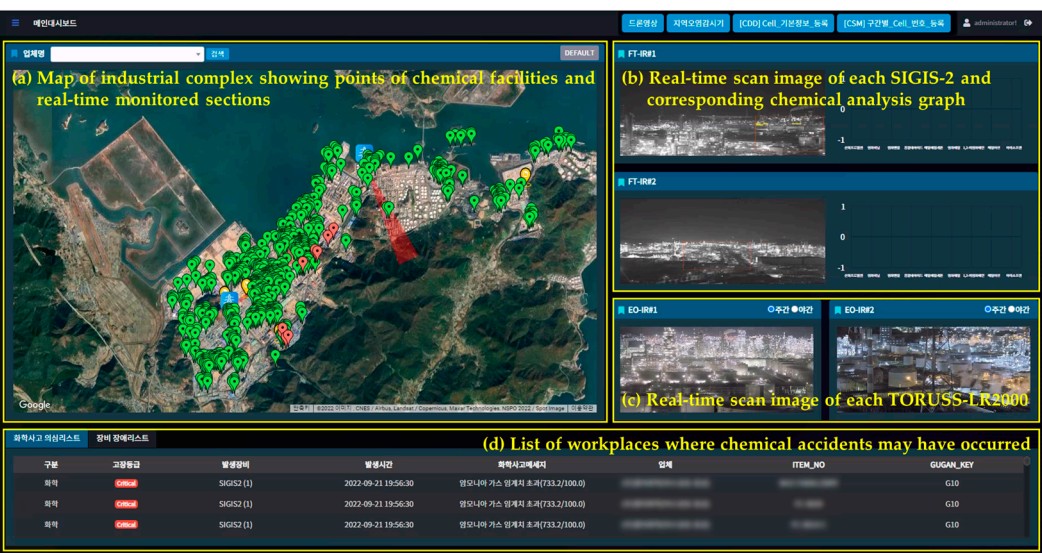

**Figure 8.** Main screen of the monitoring software. (a) Map of industrial complex showing points of chemical facilities and real-time monitored sections. (b) Real-time scan image of each SIGIS-2 and corresponding chemical analysis graph. (c) Real-time scan image of each TORUSS-LR2000. (d) List of workplaces where chemical accidents may have occurred.

In the AI chemical accident detection model, the machine learning model, upon request through REST API, detects fires, smoke, and anomalies. This model comprises an object detection model and an anomaly detection model. It transmits prediction results, and the sensitivity of the model can be adjusted by modifying the threshold for each request. An object detection model is less affected by changes in the surrounding environment, while an anomaly detection model can be sensitive to changes (such as geographic features, products in the workplace).

The Mask RCNN model [27] was applied to the object detection model in order to find the patterns of an object from features (such as components and colors) in a video or image. The object detection model detects fires or smoke through EO imaging for each section photographed by the TORUSS-LR2000. Currently, we are collecting the learning data by masking objects that are falsely detected as smoke (e.g., white storage tanks).

An autoencoder structure [28] was applied to the anomaly detection mode. This is a neural network that automatically learns useful functions and expressions from obtained data. The anomaly detection model detects anomalies (i.e., accidents) by comprehensively reviewing images, videos, and meteorological data obtained by monitoring an area. Currently, the data are pre-processed and entered into the anomaly detection model to classify the anomaly data. In the future, after sufficient data from anomalies have been collected, we will switch to supervised learning and apply various learning approaches.

## 3. Results

A test was performed to ascertain the effectiveness of the industrial complex chemical accident monitoring system. The following data were obtained for the test bed of the YNIC from July to September.

### 3.1. Chemicals Detected at Each Site

First, eight types of chemical substances were detected by the SIGIS-2 at the Jeongnyang site in 17,756 cases, including 1,3-butadiene (58 cases), dimethylformamide (1 case), methanol (3,763 cases), methylamine (940 cases), benzene (11,435 cases), ethyl acetate (43 cases), ammonia (1,487 cases), and vinyl chloride (29 cases). Figure 9(a) presents the chemical substances detected in the 13 sections at the Jeongnyang site. The SIGIS-2 at the Hwachi site detected 2,278 cases of ten different types of chemical substances, including 1,2-dichloroethane (19 cases), 1,3-butadiene (278 cases), p-xylene (19 cases), formic acid (1 case), methanol (27 cases), methylamine (842 cases), ethylene oxide (7 cases), ethyl acetate (233 cases), ammonia (40 cases), and vinyl chloride (812 cases). Figure 9(b) shows the chemical substance detection results for each of 15 out of 20 sections at the Hwachi site.

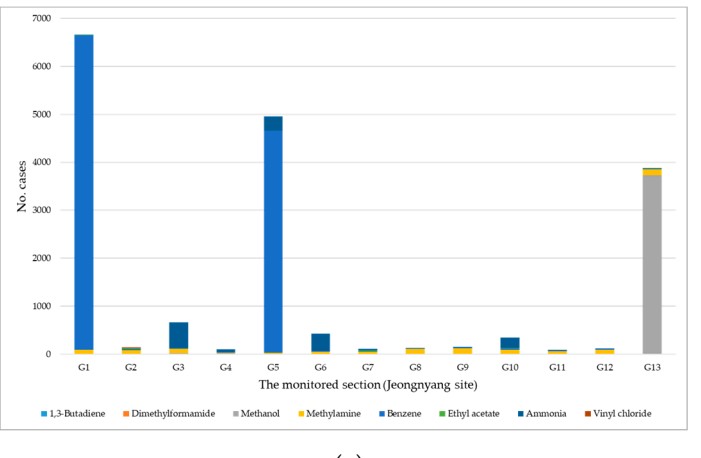

(a)

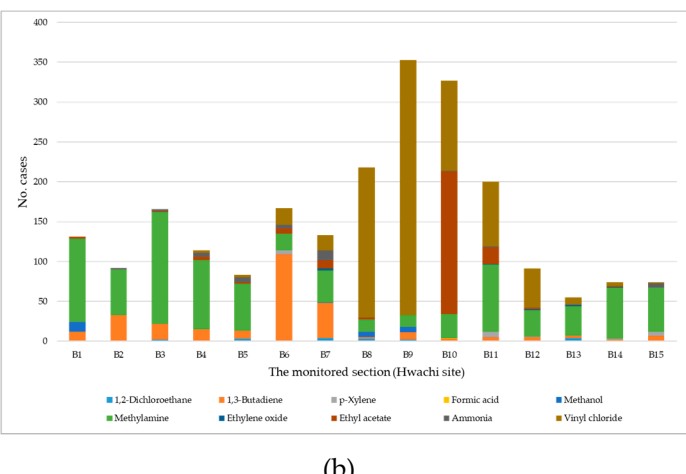

(b)

**Figure 9.** Result of each chemical substance detected in the monitored section at the (**a**) Jeongnyang site, and (**b**) Hwachi site.

### 3.2. Match the Detection Result with the Industrial Complex Information

Figure 10 shows the detection results of the SIGIS-2 for vinyl chloride in the monitored section G2, and Figure 11 shows the delta T and the calculated result of the spectrum of vinyl chloride detected by the SIGIS-2 analysis. Figure 11 shows that the detection result was almost identical to the reference spectrum of vinyl chloride in the SIGIS-2 library.

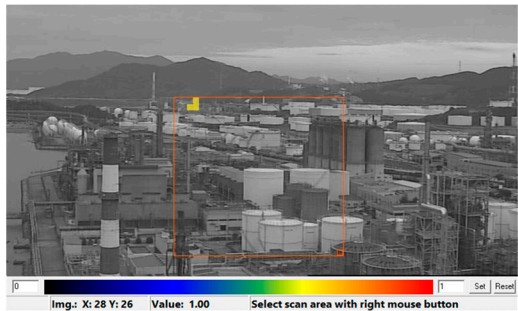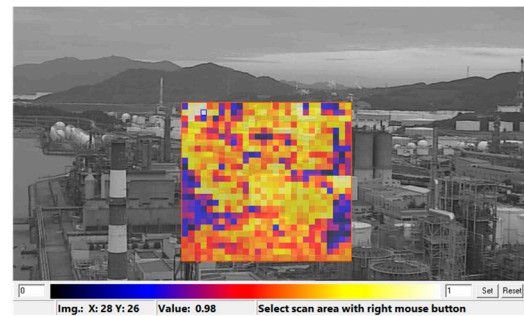

**Figure 10.** The identification image (**left**) and the coefficient of correlation image (**right**) for a case in which vinyl chloride was detected by the SIGIS-2 installed at the Jeongnyang site. (Date: 20 Sep 2022 at 18:07 GMT+9).

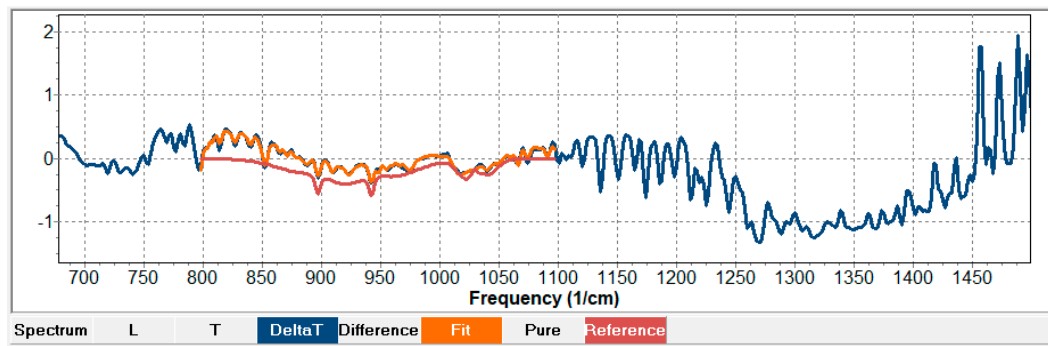

**Figure 11.** Delta T and the calculated result of the spectrum of vinyl chloride detected by the SIGIS-2 analysis.

The data used to detect vinyl chloride in section G2 are consistent with the actual handling information of vinyl chloride at the workplace, which is 2.2 km away from the steel tower at the Jeongnyang site in Figure 12.

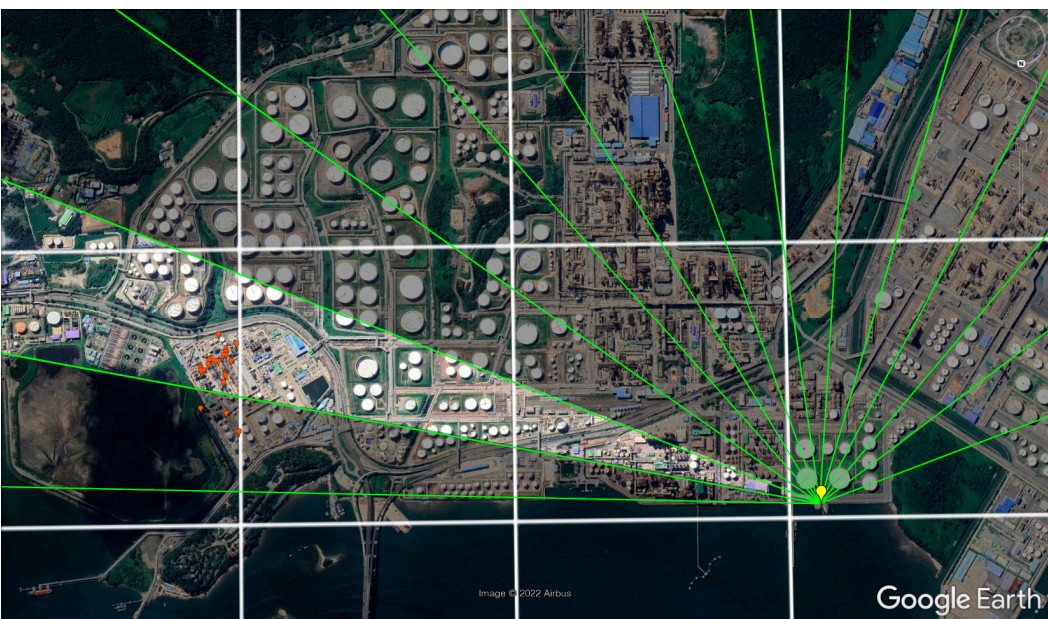

**Figure 12.** The actual handling information of vinyl chloride (orange dots) in monitoring section G2. (Source: Google earth).

The overall results confirm that the industrial complex chemical accident monitoring system, based on a SIGIS-2 installed on a 70m-high steel tower, can detect chemical substances from a distance. In Korea, when chemical accidents occur, any person who handles relevant chemicals can immediately (within 15 minutes) report the occurrence of such chemical accidents to the competent local government, local environmental agency, national police agency, fire agency, or local employment and labor agency [29]. As such, chemical accident reports can be delayed depending on the speed of the people reporting, but this system can automatically alarm chemical accidents within 5 minutes after detecting chemical leakages, fires or smoke. Therefore, it is effective for early detection. The accumulated detection data shows that the detection frequency of a specific chemical substance is high in each section. This can be used as a basis for modifying the threshold of the anomaly detection model. Additionally, it can identify chemical substances that are routinely emitted from workplaces, and based on this, the government can manage and supervise workplaces in industrial complexes.

## 4. Conclusions

The industrial complex chemical accident monitoring system was constructed. It uses an IR-RCD and an LRVS to detect chemical leakages, fires, and smoke in industrial complexes. The facilities comprise a steel tower with monitoring equipment installed at its top (thereby enabling long-range visibility), a steel tower control room that supplies power to the equipment, and a central control room for integrated monitoring and coordination. The industrial complex information includes information about the actual handled chemical substances and monitored sections. The software solutions comprise control software and an AI chemical accident detection model.

This system combines different technologies and can detect potential chemical accidents (such as chemical leakages and fire outbreaks) occurring in a specific section. As it is a fixed system with equipment mounted on the tower, it can scan the industrial complex 24 hours per day, 365 days per year. The main goal of this system is to ensure reliability by cross-checking the detected results for each piece of equipment and verifying data about industrial complexes. Moreover, the accuracy can be improved by continuously updating the information.

According to the results of a test at the YNIC, the data obtained through the SIGIS-2 were consistent with the chemical substance handling information established in the system. These results confirm that it is possible to monitor an industrial complex from a 70 m high steel tower. However, to secure reliability of the detection results, the information about the chemical substances and handling facilities at the sites must be continuously updated. We plan to investigate different applications of the TORUSS-LR2000 and the AI chemical accident detection model. This system can detect and evaluate chemical leakages and the occurrence of fires or smoke through large-scale scans, and it will be used for the early detection of and effective responses to chemical accidents in industrial complexes.

**Author Contributions:** Conceptualization, B.C.M. and S.G.L.; methodology, S.G.L.; investigation, S.G.L. and E.H.K.; writing—original draft, S.G.L. and E.H.K.; writing—review and editing, B.C.M.; supervision, B.C.M. All authors have read and agreed to the published version of the manuscript.

**Funding:** This research was supported by The Graduate School of Chemical Characterization hosted by the Korean Ministry of Environment: 2022030903900.

**Institutional Review Board Statement:** Not applicable.

**Informed Consent Statement:** Not applicable.

**Acknowledgments:** This research was conducted with the support of the Graduated School of Chemical Characterization hosted by the Korean Ministry of Environment.

**Conflicts of Interest:** The authors declare no conflict of interest.

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
