# Peer review of "Monitoring Chemical Accidents in Industrial Complexes Using Tower-Installed Infrared System for Remote Chemical Detection and Long-Range Video Surveillance System"

_applsci, doi:10.3390/app13031544_

Round 1

Reviewer 1 Report

1. In the first part of the introduction of the paper, the four contributions of the paper are summarized, but the paper does not see how to reflect these contributions,There is no valid proof.

2. Technical highlights of this paper, especially how dangerous goods are monitored by infrared technology and what technological innovation is in it, are not reflected in this paper.

3, the paper is similar to a product description, lack of necessary innovative refining. For example, what are the advantages of this technology system compared with other technology systems? Whether there are similar products and what are the core highlights of this technology are not mentioned, which need to be supplemented and perfected

Reviewer 2 Report

Dear Authors, 

I would like to express my appreciation for your submission of the manuscript, which aims to monitor chemical accidents through the use of various tools including infrared and long-range video systems.

Upon review, I found that the overall design of the manuscript is well thought-out. However, there are some areas that may require improvement. In particular, the language used in the manuscript would benefit from the expertise of a native English speaker. Additionally, there is a lack of information regarding the programming language used for the AI-based detections and the details of the newly developed software. This information would be of interest to readers and the scientific community.

I suggest that your manuscript would be a good contribution to the literature after minor revisions.

Best Regards

Round 2

Reviewer 1 Report

I think the author carefully revised the paper, the quality of the paper has been improved to a certain extent, and the paper has met the requirements for publication.